# Spatial and Temporal Diversity of Astrocyte Phenotypes in Spinocerebellar Ataxia Type 1 Mice

**DOI:** 10.3390/cells11203323

**Published:** 2022-10-21

**Authors:** Juao-Guilherme Rosa, Katherine Hamel, Carrie Sheeler, Ella Borgenheimer, Stephen Gilliat, Alyssa Soles, Ferris J. Ghannoum, Kaelin Sbrocco, Hillary P. Handler, Orion Rainwater, Ryan Kang, Marija Cvetanovic

**Affiliations:** 1Cvetanovic Laboratory, Department of Neuroscience, College of Medicine, University of Minnesota, 2101 6th Street SE, Minneapolis, MN 55455, USA; 2Graduate Program for Neuroscience, Boston University, 700 Albany Street W516, Boston, MA 02118, USA; 3Baylor College of Medicine, 1 Baylor Plaza, Houston, TX 77030, USA; 4Institute for Translational Neuroscience, College of Medicine, University of Minnesota, 2101 6th Street SE, Minneapolis, MN 55455, USA; 5Department of Lab. Medicine and Pathology, College of Medicine, University of Minnesota, 2101 6th Street SE, Minneapolis, MN 55455, USA; 6Bowdoin College, Brunswick, ME 04011, USA

**Keywords:** astrocytes, SCA1, neurodegeneration, microglia, morphology, gene expression, brain region

## Abstract

While astrocyte heterogeneity is an important feature of the healthy brain, less is understood about spatiotemporal heterogeneity of astrocytes in brain disease. Spinocerebellar ataxia type 1 (SCA1) is a progressive neurodegenerative disease caused by a CAG repeat expansion in the gene *Ataxin1* (*ATXN1*). We characterized astrocytes across disease progression in the four clinically relevant brain regions, cerebellum, brainstem, hippocampus, and motor cortex, of *Atxn1^154Q/2Q^* mice, a knock-in mouse model of SCA1. We found brain region-specific changes in astrocyte density and GFAP expression and area, early in the disease and prior to neuronal loss. Expression of astrocytic core homeostatic genes was also altered in a brain region-specific manner and correlated with neuronal activity, indicating that astrocytes may compensate or exacerbate neuronal dysfunction. Late in disease, expression of astrocytic homeostatic genes was reduced in all four brain regions, indicating loss of astrocyte functions. We observed no obvious correlation between spatiotemporal changes in microglia and spatiotemporal astrocyte alterations, indicating a complex orchestration of glial phenotypes in disease. These results support spatiotemporal diversity of glial phenotypes as an important feature of the brain disease that may contribute to SCA1 pathogenesis in a brain region and disease stage-specific manner.

## 1. Introduction

Astrocytes play critical roles in brain development and functioning of the adult brain [1,2,3]. One key way astrocytes regulate neuronal activity is by maintaining homeostasis of ions, neurotransmitters, and water in the brain [4,5,6,7]. Differences in the activity and metabolic needs of neurons across and within brain regions indicate the heterogeneous and region-specific activity of astrocytes [8,9,10,11,12,13,14]. While heterogeneity of astrocyte morphology was described by Ramon Y Cajal, recent advances in single-cell RNA sequencing (scRNA-seq), spatial transcriptomics, and single-cell mapping of astrocyte interactions have begun characterizing the molecular heterogeneity of astrocytes and its relevance for healthy brain function [15,16,17].

In brain diseases, astrocytes undergo a phenotypic change termed reactive gliosis, defined as morphological and gene expression changes that may result in altered astrocyte function [18]. Reactive astrocytes contribute to pathogenesis of neurodegenerative disease [19], and can either exacerbate or ameliorate neurodegeneration as has been shown in Huntington’s disease (HD) [20,21], Alzheimer’s disease (AD) [22,23,24], and spinocerebellar ataxia type 1 (SCA1) [25]. It is still unknown how regional heterogeneity of astrocytes affects their changes in disease and whether it contributes to the selective regional neuronal vulnerability, which is one of the key features of neurodegenerative diseases [26,27].

SCA1 is an inherited neurodegenerative disorder caused by CAG repeat expansion in the *Ataxin-1* (*ATXN1)* gene [28]. Repeat expansions of about 39 or more CAG repeats result in the expression of a toxic polyglutamine (polyQ) ATXN1 protein [29]. SCA1 is characterized by progressive degeneration that is most severe in the cerebellum and brainstem, but is also present in other brain regions, and manifests behaviorally as deficits in motor coordination (i.e., ataxia), swallowing, speaking, cognition, and mood [30,31,32]. While *Atxn1*-targeting antisense oligonucleotides (ASOs) show promise in pre-clinical trials [33], no disease-modifying treatments are currently available for SCA1.

Motor deficits and underlying cerebellar pathology have been the focus of the majority of SCA1 studies so far. However, *ATXN1* is expressed throughout the brain and likely affects brain regions beyond the cerebellum [34,35]. ATXN1 protein is thought to function as a transcriptional regulator as it interacts with several transcription factors, and gene expression changes are one of the earliest alterations in SCA1 mice. Pathology is described in the brainstem, motor cortex, and hippocampus of SCA1 patients, and is thought to contribute to SCA1 symptoms such as cognitive deficits, mood disorders, difficulties in respiration and swallowing, and premature lethality [36,37,38,39,40,41]. We have previously demonstrated that astrocytes actively contribute to disease pathogenesis in the cerebellum of SCA1 mice [25], but how astrocytes are altered in other SCA1-affected brain regions remains unknown.

To better understand brain-wide SCA1 astrocyte pathology, we have utilized the *Atxn1^154Q/2Q^* mouse model of SCA1 wherein 154 CAG repeats have been knocked-in into the endogenous mouse *Atxn1* gene [42]. Because the *Atxn1* gene is expressed under the endogenous promoter, *Atxn1^154Q/2Q^* mice express polyQ ATXN1 in cells throughout the central nervous system [https://www.proteinatlas.org/, accessed on March 2022] and in brain cells such as neurons, astrocytes, and microglia [http://www.brainrnaseq.org/, accessed on March 2022]. As a result, in addition to motor deficits, the *Atxn1^154Q/2Q^* mice exhibit premature lethality, failure to gain weight, and cognitive deficits [42,43]. Furthermore, there is evidence of hippocampal and brainstem changes in *Atxn1^154Q/2Q^* mice, although pathology in these brain regions has yet to be thoroughly characterized [42,44,45].

To investigate the region-specific and temporal changes in SCA1 astrocytes, we compared morphology and gene expression changes in astrocytes in the cerebellum, brainstem, hippocampus, and motor cortex of 12- and 20-week-old *Atxn1^154Q/2Q^* mice. We focused on these time points because they represent early- to mid- and late-disease stages. Early stages of disease are particularly of interest due to previous studies showing the significant potential of early therapeutic approaches [46,47].

We determined astrocytic density, area, and GFAP expression, and the expression of core astrocytic homeostatic genes including *Kcnj10*, *Aqp4*, *Slc1a2,* and *Gja1*. These astrocytic genes play a critical role in the maintenance of brain homeostasis by regulating local concentrations of potassium (*Kcnj10*), water (*Aqp4*), and glutamate (*Slc1a2*) [7,48,49,50].

We also investigated microglial and neuronal changes across these four brain regions. We examined changes in microglial density and area, and quantified neuronal density, activity, and loss of excitatory synapses in the examined regions.

## 2. Materials and Methods

### 2.1. Mice

The creation of the *Atxn1^154Q/2Q^* mice was previously described [42]. Because repeat length in trinucleotide repeat expansions is unstable and prone to further expansion [51], we routinely perform genetic sequencing of our mouse lines. We found that the number of repeats has recently expanded in our colony from 154 CAG to 166 CAG. Based on previous studies, this increase in CAG repeat number is expected to increase the severity of disease and lower the age of symptom onset. We used an approximately equal number of male and female mice in all experiments.

### 2.2. Immunofluorescent (IF) Staining

IF was performed on a minimum of six different floating 45-μm-thick brain slices from each mouse (six technical replicates per mouse per region or antibody of interest). Confocal images were acquired using a confocal microscope Olympus FV1000 (Leica, Deerfield, IL, USA) using a 20×, 40×, or 60× oil objective. Z-stacks consisting of twenty non-overlapping 1-μm-thick slices were taken of each stained brain slice per brain region (i.e., six z-stacks per mouse, each taken from a different brain slice). The laser power and detector gain were standardized and fixed between mice within a surgery cohort, and all images for mice within a cohort were acquired in a single imaging session to allow for quantitative comparison.

We used primary antibodies against Purkinje cell marker calbindin (mouse, Sigma-Aldrich, C9848, St. Louis, MO, USA), neuronal marker neuronal nuclei (NeuN) (rabbit, Ab104225, Abcam, Boston, MN, USA), astrocytic marker glial fibrillary acidic protein (GFAP) (chicken, AB5541, Millipore, Madison, WI, USA), microglial marker ionized calcium binding adaptor molecule 1 (Iba1) (rabbit, 019-19741, WAKO, Richmond, VA, USA), c-Fos (rabbit, ab190289, Abcam, Boston, MN, USA), PSD-95 (mouse, 810401, BioLegend, San Diego, CA, USA), and vesicular glutamate transporter 2 (VGLUT2) (guinea pig, AB2251-I, Millipore, Madison, WI, USA) as previously described [25,52]. We used specie-specific corresponding secondary antibodies conjugated with Alexa 488 and Alexa 594 (Jackson Immunoresearch Laboratories, West Grove, PA, USA). Quantitative analysis was performed using ImageJ (NIH, Bethesda, MA, USA). To quantify relative intensity of staining for GFAP and calbindin, we measured the average signal intensity in the region of interest and normalized it to that of the WT littermate controls. The density of neurons, astrocytes, microglia, and active neuronal cells was determined by normalizing the number of NeuN+, GFAP+, Iba1+, and c-Fos+ cells, respectively, with the area of the region of interest. We determined GFAP+ and Iba1+ percent area by creating a mask of GFAP and Iba1 staining, respectively, and recording the fraction of the region of interest covered by staining. To quantify atrophy of the cerebellar molecular layer, we took six measurements per image of the distance from the base of the Purkinje soma to the end of their dendrites, the average being the molecular layer width for that image. Numbers of VGLUT2 and PSD95 puncta, and their co-localizations, were quantified on 60× 4× zoomed images using Puncta Analyzer Plugin (Durham, NC, USA) in ImageJ. This plugin takes the advantage of the fact that presynaptic (VGLUT2) and postsynaptic (PSD95) proteins reside in separate cell compartments, but because of their close proximity they appear to co-localize at synapses. Puncta Analyzer Plugin is set to subtract background with rolling ball radius of 50 pixels. Threshold was selected for each image and channel to further remove any potential background noise, and particle size was set at 4 pixels. At least 3 optical sections per brain slice, and at least 2 brain slices per animal were analyzed, making a total of 16–27 image data sets per brain region for each genotype. The width of hippocampal neuronal layers (CA2 and CA3) was measured by dividing the area of each neuronal layer by the length of that layer as previously described [44].

### 2.3. RNA Extraction, Sequencing, and Analyses

Cerebellum, medulla, cerebral cortex, and hippocampal tissue was isolated from 26-week-old wild-type and *Atxn1^154Q/2Q^* mice and stored in RNAlater solution (Thermo Fisher Scientific, Coon Rapids, MN, USA). Total RNA was isolated using TRIzol reagent (Thermo Fisher Scientific, Coon Rapids, MN, USA) following the manufacturer’s protocols. Tissue was homogenized using RNase-Free disposable pellet pestles in a motorized chuck. Purified RNA was sent to the University of Minnesota Genomics Center for quality control, including quantification using fluorimetry via RiboGreen assay kit (Thermo Fisher Scientific, Coon Rapids, MN, USA), and RNA integrity was assessed via capillary 34 electrophoresis using an Agilent BioAnalyzer 2100 to generate an RNA integrity number (RIN). RIN values for submitted RNA were above 8.0 for all samples except one medulla sample (RIN = 6.8). All submitted RNA samples had greater than 1µg total mass. Library creation was completed using oligo-dT purification of polyadenylated RNA, which was reverse transcribed to create cDNA. cDNA was fragmented, blunt-ended, and ligated to barcode adaptors. Libraries were size selected to 320 bp ± 5% to produce average inserts of approximately 200 bp, and size distribution was validated using capillary electrophoresis and quantified using fluorimetry (PicoGreen, Thermo Fisher Scientific, Coon Rapids, MN, USA) and qPCR. Libraries were then normalized, pooled, and sequenced on an S4 flow cell by an Illumina NovaSeq 6000 using a 150-nucleotide, paired-end read strategy. The resulting FASTQ files were trimmed, aligned to the mouse reference genome (GRCm38), sorted, and counted using the Bulk RNAseq Analysis Pipeline from the Minnesota Supercomputing Institute’s Collection of Hierarchical UMII/RIS Pipelines (v0.2.0). Genes less than 300 bp are too small to be accurately captured in standard RNAseq library preparations, so they were discarded from all downstream analyses.

Differential gene expression analysis was performed using the edgeR package (v3.30.3) in R (R Foundation for Statistical Computing v3.6.1). All four brain regions were analyzed independently. Genes with fewer than 10 counts across all samples in each region were excluded. Genes with FDR values less than or equal to 0.05 were considered significant.

### 2.4. Reverse Transcription and Quantitative Polymerase Chain Reaction (RT-qPCR)

Total RNA was extracted from dissected mouse cerebella, brainstem, cortex, and hippocampus using TRIzol (hermo Fisher Scientific, Coon Rapids, MN, USA), and RT-qPCR was performed as described previously [8]. We used IDT Primetime primers (IDT, Coralville, IW, USA). Relative mRNA levels were calculated using 18S RNA as a control and wild-type mice as a reference using 2^−ΔΔCt^ as previously described [8].

### 2.5. Statistics

Wherever possible, sample sizes were calculated using power analyses based on the standard deviations from our previous studies, significance level of 5%, and power of 90%. Statistical tests were performed with GraphPad Prism 7.0 (GraphPad Software, San Diego, CA, USA). Data was analyzed using two-way ANOVA (to assess the impact of genotype and treatment) followed by post-hoc two-tailed, unpaired Student’s *t*-test (to test statistical significance of differences between groups of interest wherein only one variable was different (treatment or genotype), or one-way ANOVA followed by the Sidak’s post-hoc test. Outliers were determined using GraphPad PRISM’s Robust regression and Outlier removal (ROUT) with a Q = 1% for non-biased selection.

## 3. Results

### 3.1. Spatial Diversity of Astrocyte Morphology during Early Disease Stages in Atxn1^154Q/2q^ Mice

Ataxin-1 is widely expressed throughout the brain in humans and in mice [44,53,54] yet how astrocytes are altered across relevant brain regions in SCA1 is unknown [35]. We first investigated morphological changes in astrocytes in the cerebellum, brainstem, hippocampus, and motor cortex of *Atxn1^154Q/2Q^* mice during early disease stage. These regions have presumptive roles in motor deficits, premature lethality, and cognitive dysfunction in SCA1. Previously described neuronal degeneration and gliosis in the brainstem of patients with SCA1 are thought to contribute to the loss of ability to protect airways and, subsequently, aspiration pneumonia and death [36,55].

To investigate morphological changes in SCA1 astrocytes, we quantified an increase in area occupied by glial fibrillary acidic protein (GFAP) (percentage of area that is GFAP+), which is a morphological characteristic of hypertrophy in reactive astrocytes, an increase in GFAP immunoreactivity, which is often used as a measure of reactive astrogliosis, and cell density where possible [56,57]. In the cerebellum, we examined Bergmann glia (BG), a subtype of cerebellar astrocytes that reside in the Purkinje and molecular layers and have a very intimate structural and functional relationship with Purkinje cells (PCs), neurons affected in SCA1 [58,59]. We detected a significant increase in GFAP+ percent area (Figure 1A), and in the intensity of GFAP staining [60], indicative of hypertrophy of SCA1 Bergmann glia. We have previously described SCA1 alterations in the dentate gyrus of the hippocampus in *Atxn1^154Q/2Q^* mice [45]. We examined astrocyte morphology and found an increase in intensity of GFAP expression and GFAP+ astrocyte percent area, indicative of astrocyte hypertrophy in the dentate gyrus of *Atxn1^154Q/2Q^* mice relative to wild-type controls (Figure 1B). In the brainstem, we characterized astrocyte morphology in the inferior olivary nucleus (ION) [41], because postmortem analysis of patient samples showed severe loss of volume in the ION [34], and due to the high-quality of GFAP staining relative to other brainstem nuclei (Figure 1C). Interestingly, we found that in the ION of *Atxn1^154Q/2Q^* mice, the intensity of GFAP staining and density of astrocytes were significantly reduced. Previous brain-wide investigation demonstrated neuronal loss in the primary motor cortex of SCA1 patients [34]. Because of the reported differences in gray and white matter astrocytes, we assessed astrocyte morphology in layer 6 of the motor cortex and the underlying corpus callosum of *Atxn1^154Q/2Q^* mice (Figure 1D). We detected a slight increase in GFAP intensity and percent area in the corpus callosum of *Atxn1^154Q/2Q^* mice that did not reach statistical significance and may indicate emerging hypertrophy of cortical astrocytes (Figure 1D). There were no changes in astrocyte density, GFAP intensity, or area in the layer 6 of motor cortex.

These results indicate diverse morphological changes in SCA1 astrocytes across brain regions with astrocytic loss and atrophy in the brainstem, astrocyte hypertrophy in the cerebellum, and in the hippocampus and trending, but not significant astrocytic hypertrophy in the corpus callosum.

### 3.2. Spatial Diversity of Astrocytic Gene Expression Changes in SCA1

We next investigated molecular changes in SCA1 astrocytes across brain regions. Among the astrocytic roles critical for neuronal function is maintaining the homeostasis of the extracellular environment. Astrocytes accomplish this by removing excess extracellular glutamate via glutamate transporters encoded by the *Slc1a3* and *Slc1a2* genes, potassium ions via the potassium channel encoded by *Kcnj10*, and water through the expression of Aquaporin-4, encoded by *Aqp4* [61,62,63,64]. Both *Slc1a2* and *Slc1a3* are present in astrocytes throughout the brain, but *Slc1a3* is preferentially expressed in the cerebellum and *Slc1a2* in the hippocampus. Astrocyte intercommunication in a network allowing for calcium wave propagation, and also potassium and glutamate buffering, is facilitated by Connexin43 encoded by *Gja1*. We compared the molecular changes in SCA1 astrocytes across the cerebellum, brainstem (medulla), hippocampus, and cortex by quantifying the expression of these critical astrocytic core genes using RT-qPCR.

In the cerebellum of *Atxn1^154Q/2Q^* mice, we have found a large variability in the expression of *Aqp4*, *Gja1*, *Kcnj10*, and *Slc1a3*, and no significant change compared to wild-type controls (Figure 2A). We found a significant reduction in the expression of *Aqp4*, *Gja1*, *Slc1a2*, and *Kcnj10*, in the hippocampus, and with the exception of *Slc1a2*, also in the brainstem (Figure 2B,C) of *Atxn1^154Q/2Q^* mice at 12 weeks. In contrast, expression of these genes was increased in the cortex (Figure 2D). Given the importance of these homeostatic astrocytic genes for neuronal function, our results may indicate a novel mechanism by which astrocytes exacerbate neuronal dysfunction in the brainstem and hippocampus and compensate for, or ameliorate dysfunction in, the cortex of *Atxn1^154Q/2Q^* mice.

The large variability in the expression of *Aqp4*, *Gja1*, *Kcnj10*, and *Slc1a3*, in the cerebellum may reflect large variability in the reactive state of cerebellar astrocytes (as determined by the *Gfap* expression). Indeed, we found a positive correlation between the expression of *Aqp4*, *Gja1*, and *Slc1a3*, and the expression of *Gfap* (Appendix A), and a trending correlation between *Gfap* and *Kcnj10* (data not shown). These results indicate that early in disease, cerebellar astrocytes with a higher *Gfap* expression (denoting that they are likely to be more reactive) also have a higher expression of homeostatic genes.

### 3.3. Regional Diversity of Microglial SCA1 Changes

We have previously shown that microglia are reactive in SCA1 cerebellar cortex and contribute to SCA1 pathogenesis in different SCA1 mouse models [65,66]. We investigated changes in microglial density and morphology in the cerebellum and other brain regions using IHC staining for the microglial marker ionized calcium-binding adaptor protein (Iba1). We quantified microglial density and the percent area covered by microglial Iba1+ processes (Figure 3). In the cerebellum, we found a significant increase in microglia density and Iba1+ percent area in 12-week-old *Atxn1^154Q/2Q^* mice (Figure 3A). Similarly, in the ION (Figure 3C) and cortex (Figure 3D), we found an increase in microglial density. Microglial area, indicative of hypertrophy, was significantly increased only in the cortex. We found no significant changes in microglial density or percent area in the hippocampus (Figure 3B).

Reactive changes in glia can be caused by neuronal loss. To determine whether observed glial changes may be caused by neuronal loss, we used neuronal marker NeuN to quantify neuronal numbers. We did not find evidence of neuronal loss in any of the examined brain regions at this early disease stage in *Atxn1^154Q/2Q^* mice (Appendix A). While this result does not exclude neuronal dysfunction, our results suggest that spatially diverse changes in glial phenotypes precede neuronal loss.

To investigate whether observed changes in the glial phenotypes may be associated with more subtle neuronal changes, we quantified neuronal activity in the hippocampus and cortex, two regions with reduced and increased expression of core astrocyte genes, respectively. Intriguingly, the ratio of cFos+ neurons to total NeuN+ neurons was decreased in the hippocampus while it was not changed in the motor cortex of *Atxn1^154Q/2Q^* mice (Appendix A).

Together, these results suggest that microglia—similar to astrocytes—exhibit spatial diversity of SCA1 changes. Intriguingly, while early in disease astrocytes undergo both loss and hypertrophy, microglia mostly undergo hypertrophy, indicating a complex orchestration of astrocyte and microglial phenotypes early in SCA1. Moreover, decreased astrocytic support may contribute to reduced neuronal activity in the hippocampus.

### 3.4. Temporal Diversity of Astrocyte Phenotypes in SCA1s

We next investigated how glial phenotypes change with disease progression. At late stage of disease (20 weeks), astrocytes in the cerebellum, hippocampus, and motor cortex showed signs of hypertrophy (Figure 4A,B,D). In sharp contrast, in the inferior olivary nucleus (ION) both astrocyte density and area occupied markedly declined (Figure 4C).

Using RNA sequencing, we compared the expression of astrocytic genes in these regions of *Atxn1^154Q/2Q^* mice and their wild-type littermate controls. We found a significant decrease in the expression of astrocytic homeostatic genes *Kcnj10*, *Aqp4*, *Gja1*, and *Slc1a2* in all four regions in the late stage *Atxn1^154Q/2Q^* mice (Figure 5). These results indicate temporal diversity of astrocyte phenotypes in SCA1 and suggest that brain-wide loss of astrocyte core gene expression characterizes late stages of disease in *Atxn1^154Q/2Q^* mice.

Intriguingly, late in disease both the ION and hippocampus demonstrated a decrease in microglial density and in microglia area (although decrease in the microglia area was only trending in the hippocampus) (Figure 6). While density and area of cerebellar microglia was further increased with disease progression, we did not detect any change in microglial density or area in the cortex.

Microglia have been suggested to lead to loss of synapses in disease. To investigate whether changes in microglia density are associated with loss of synapses, we quantified the number of excitatory synapses in brain regions with different changes in microglia: the cerebellum that exhibits microglial activation, and the cortex and hippocampus that respectively show reduction, or no change, in microglial density. We found significant reduction of VGLUT2/PSD95 synapses in the cerebellum, while there was no change in the VGLUT2/PSD95 synaptic quanta in the hippocampus and in the motor cortex (Appendix A).

These results suggest that with disease progression, some brain regions (such as olives) may suffer from loss of both astrocytes and microglia, while other regions, such as the cerebellum, may suffer from reactive phenotypes in both cell types.

## 4. Discussion

Here, we report on the spectrum of reactive astrocyte phenotypes, across several clinically relevant brain regions of *Atxn1^154Q/2Q^* mice and at different stages of disease progression summarized in Table 1. Recent studies indicated heterogeneity of glial cells across healthy brain regions [16] and across different disease conditions [56,67]. Our results indicate spatial diversity of morphological and molecular changes in astrocytes during the early stages of disease in *Atxn1^154Q/2Q^* mice. This spectrum of reactive astrocytic phenotypes may indicate brain region-specific astrocytic dysfunctions and consequently their brain region-specific contributions to SCA1 disease pathogenesis. Given that these changes precede neuronal loss, it is crucial to examine glial heterogeneity as a means to distinguish protective versus deleterious astrocytic phenotypes and identify opportunities for early therapeutic intervention. For this reason, we focused on the astrocytic neuroprotective genes *Slc1a2*, *Kcnj10*, *Gja1*, and *Aqp4*, and found that their expression was reduced in the medulla and hippocampus, but was increased in the cortex of *Atxn1^154Q/2Q^* mice at 12 weeks of age and before notable neuronal loss [6]. *Kcnj10* encodes a potassium rectifier, Kir4.1, that is involved in maintaining potassium homeostasis [68] and has been implicated in several diseases, including Huntington’s disease, ALS, and depression [68,69]. *Gja1* encodes the astrocyte gap-junction protein connexin 43 (Gja1), which is critical for astrocyte–astrocyte communication [50]. *Aquaporin* (*Aqp4*) is critical for both interstitial fluid drainage in the glymphatic system and water homeostasis [70]. Astrocytic *Slc1a* (2 and 3) genes encode glutamate transporters that are responsible for removing glutamate from synaptic space. These astrocytic genes are critical for neuronal function and have been implicated in the pathogenesis of several neurodegenerative diseases [71,72]. We propose that reduced expression of astrocyte neuro-supportive genes—as observed in *Atxn1^154Q/2Q^* mice—may exacerbate neuronal dysfunction and SCA1 pathogenesis in the hippocampus and medulla. Conversely, increased expression of these neuro-supportive astrocyte genes in the cortex may indicate compensatory roles of astrocytes that delay neuronal dysfunction in the SCA1-affected cortex. This is supported by our results showing preserved neuronal activity in the cortex and reduced neuronal activity in the hippocampus.

Early in disease, we found no evidence of increased GFAP expression, a hallmark of reactive astrogliosis, in the medulla. Likewise, we found only trending indications of increased GFAP expression in the cortex. Thus, our results may indicate that increased GFAP expression may not negatively correlate to changes in astrocyte homeostatic gene expression in disease conditions. This is consistent with a previous study which demonstrated that reduced expression of the astrocyte homeostatic gene *Kcnj10* in the striatum of Huntington’s mice is not dependent on astrogliosis as defined by increased GFAP expression [68]. Moreover, in our study we have found a strong positive correlation of *Gfap* expression with the expression of *Kcnj10*, *Slc1a2,* and *Aqp4* in the cerebellum, indicating that reactive astrocytes (indicated by increased *Gfap* expression) may initially increase expression of core homeostatic genes as a neuro-supportive measure in disease.

We also found a loss of astrocytes in the ION of 12-week-old *Atxn1^154Q/2Q^* mice, preceding neuronal loss. This is consistent with previous work showing lower levels of myo-inositol (Ins), an astrocytic marker, in the brainstem of 18- and 28-week-old *Atxn1^154Q/2Q^* mice [33]. Moreover, treatment with Atxn1-targeting antisense oligonucleotides (ASO), which are also taken up by astrocytes [73], was shown to rescue lower Ins levels in the brainstem and prolong the survival of *Atxn1^154Q/2Q^* mice [33]. Based on these results, we speculate that reduced expression of homeostatic genes and decreased astrocyte viability may contribute to brainstem dysfunction and perhaps even to premature lethality in SCA1 mice. However, while neuronal numbers were not altered in the ION of 12-week-old *Atxn1^154Q/2Q^* mice, we cannot exclude neuronal dysfunction. Future studies using conditional SCA1 mice to selectively express or delete mutant ATXN1 in astrocytes will provide insight into the cell autonomous effect of mutant ATXN1 on astrocyte pathology in the ION and other brain regions in SCA1.

Finally, we report that disease progression eliminates diversity of astrocyte gene expression changes across brain regions. On the other hand, morphological changes in astrocytes became more pronounced with disease. As disease progressed, the hippocampus and brainstem exhibited a decrease in microglial density. Together, these results indicate that profound loss of glial functions characterizes late disease stage in *Atxn1^154Q/2Q^* mice and is likely to exacerbate neuronal dysfunction and loss at this stage of disease (Figure 7). While future studies will investigate mechanisms underlying the loss of astrocytes and microglia, our results indicate that preserving glial functionality may provide therapeutic benefits in SCA1.

## Figures and Tables

**Figure 1 cells-11-03323-f001:**
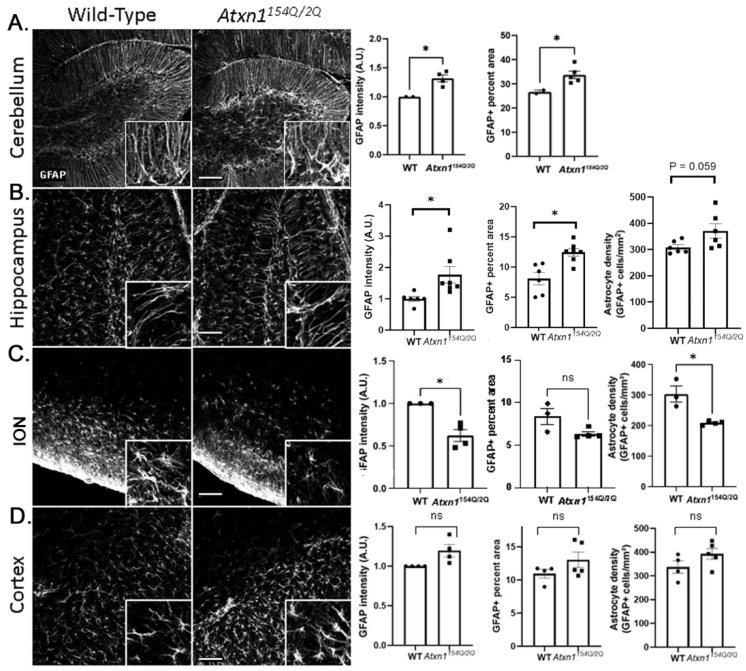
Spatial diversity of astrocytic morphology in *Atxn1^154Q/2Q^* mice early in disease. Brain sections from 12 weeks old *Atxn1^154Q/2Q^* and wild-type littermate controls were stained for GFAP. Confocal images of cerebellum ((**A**), lobule X), hippocampus ((**B**), dentate gyrus), brainstem ((**C**), ION (inferior olivary nucleus), and motor cortex ((**D**), layer 6 and corpus callosum) were used to quantify GFAP intensity, density of astrocytes, and GFAP area. Images were acquired at 20× (636.16 µm × 636.16 µm). Scale bars show 100 µm. Insets in the lower right corners of each image show zoomed images (79.5 µm × 79.5 µm) for qualitative assessment of astrocyte morphology. Data are presented as mean ± SEM with average values for each mouse represented by a dot. N = 3–7 mice of each genotype (WT and *Atxn1^154Q/2Q^*) as noted by dots representing individual mice in each bar graph. ns: not significant, * *p* < 0.05 Students *t*-test.

**Figure 2 cells-11-03323-f002:**
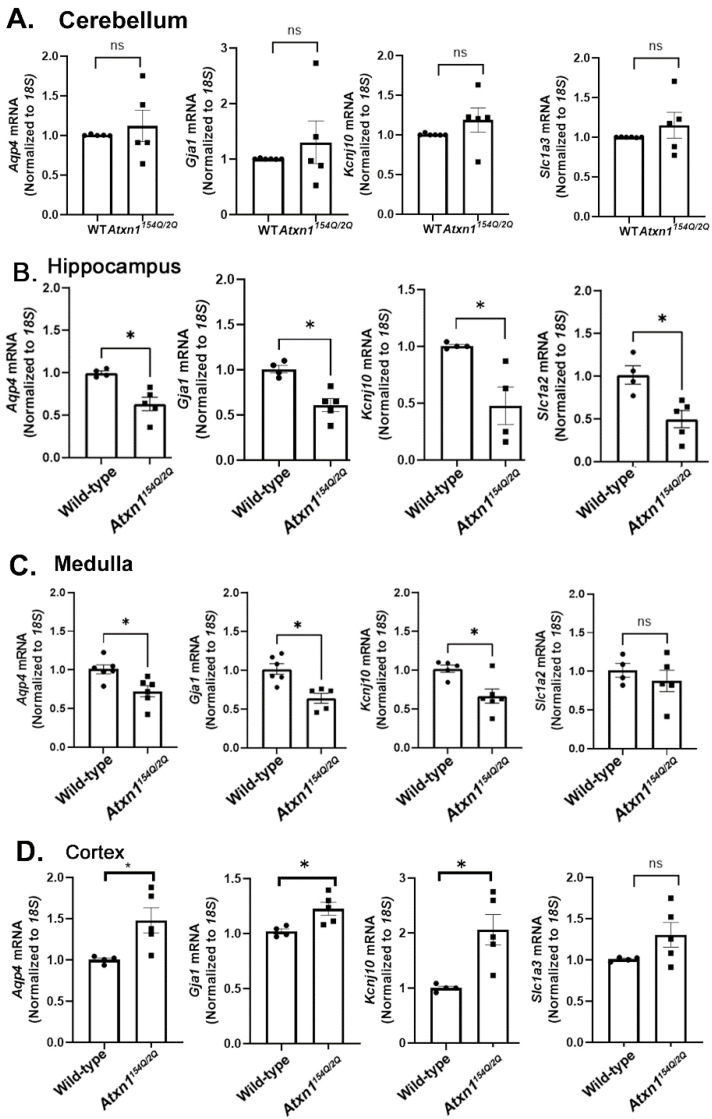
Spatial diversity of molecular changes in astrocytes during early SCA1. mRNA was extracted from the cerebellum (**A**), hippocampus (**B**), medulla (**C**), and cortex (**D**) of 12 weeks old *Atxn1^154Q/2Q^* mice and wild-type littermate controls (N = 4–6) and RTqPCR was used to evaluate expression of astrocyte specific genes. Data are presented as mean ± SEM with average values for each mouse. N = 4–6 mice of each genotype (WT and *Atxn1^154Q/2Q^*) with individual mice represented by a dot in bar graphs. ns: not significant, * *p* < 0. 05 Students *t*-test.

**Figure 3 cells-11-03323-f003:**
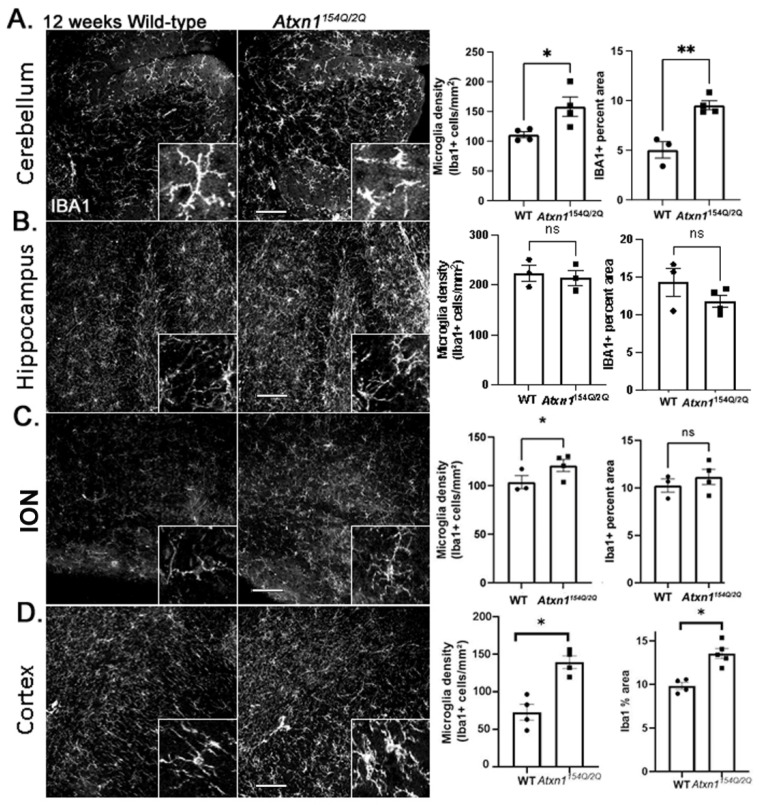
Early SCA1 microglial morphology in *Atxn1^154Q/2Q^* mice. Brain sections from 12 weeks old Atxn1154Q/2Q and wild-type littermate controls (N = 3–5) were stained for Iba1. Confocal images of cerebellum ((**A**), lobule X), hippocampus ((**B**), dentate gyrus), brainstem ((**C**), ION), and motor cortex ((**D**), layer 6 and underlying corpus callosum) were used to quantify density of Iba1 positive microglia and Iba1 area. Images were acquired at 20× (636.16 µm × 636.16 µm). Scale bars show 100 µm. Insets in the lower right corners of each image show zoomed images (79.5 × µm 79.5 µm) for qualitative assessment of microglia morphology. Data are presented as mean ± SEM with average values for each mouse. N = 3–5 mice of each genotype (WT and *Atxn1^154Q/2Q^*) with individual mice represented by a dot in bar graphs. ns: not significant, * *p* < 0.05, ** *p* < 0.01. Student’s *t*-test.

**Figure 4 cells-11-03323-f004:**
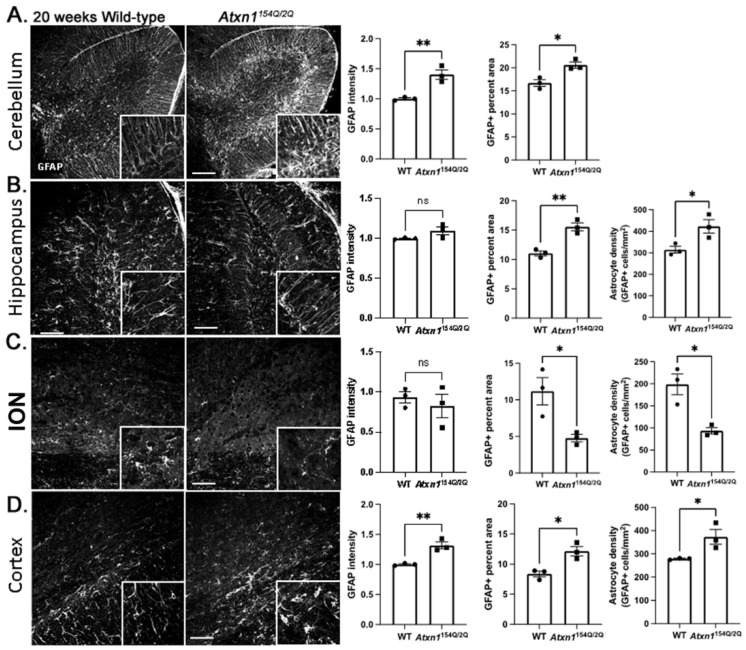
Spatial diversity of astrocytic morphology in Atxn1154Q/2Q mice at late disease stage. Brain sections from 20 weeks old Atxn1154Q/2Q and wild-type littermate controls were stained for GFAP. Confocal images of cerebellum ((**A**), lobule X), hippocampus ((**B**), dentate gyrus), brainstem ((**C**), ION), and motor cortex ((**D**), layer 6 and corpus callosum) were used to quantify GFAP intensity, density of astrocytes and GFAP area. Images were acquired at 20× (636.16 µm × 636.16 µm). Scale bars show 100 µm. Insets. in the lower right corners of each image show zoomed images (79.5 µm × 79.5 µm) for qualitative assessment of astrocyte morphology. Data are presented as mean ± SEM with average values for individual mice represented by a dot in bar graphs. N = 3–5 mice of each genotype (WT and *Atxn1^154Q/2Q^*). ns: not significant, * *p* < 0.05, ** *p* < 0.01, Students *t*-test.

**Figure 5 cells-11-03323-f005:**
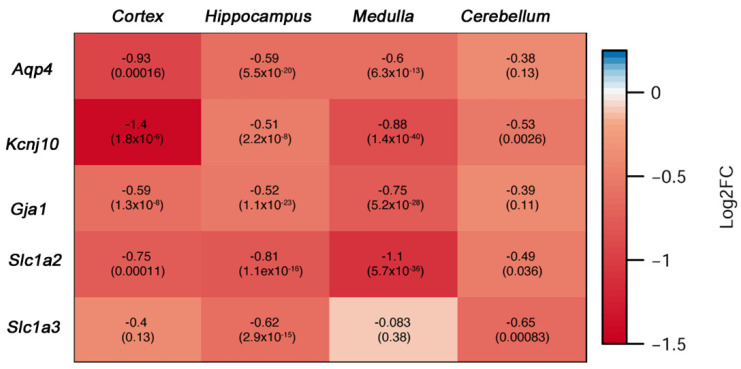
Late SCA1 stage astrocyte molecular analysis indicates brain-wide loss of core homeostatic functions. Cerebellum, brainstem (medulla), hippocampus, and cortex were isolated from 26 weeks old *Atxn1^154Q/2Q^* and wild-type control mice (N = 4 of each genotype) and isolated RNA was sent for RNA-sequencing (RNA-seq). Expression of astrocyte genes was determined using differential gene expression analysis with numbers in each square representing log2 fold change (log2FC) in *Atxn1^154Q/2Q^* compared to wild-type controls and numbers in brackets presenting False Discovery Rate adjusted *p* values.

**Figure 6 cells-11-03323-f006:**
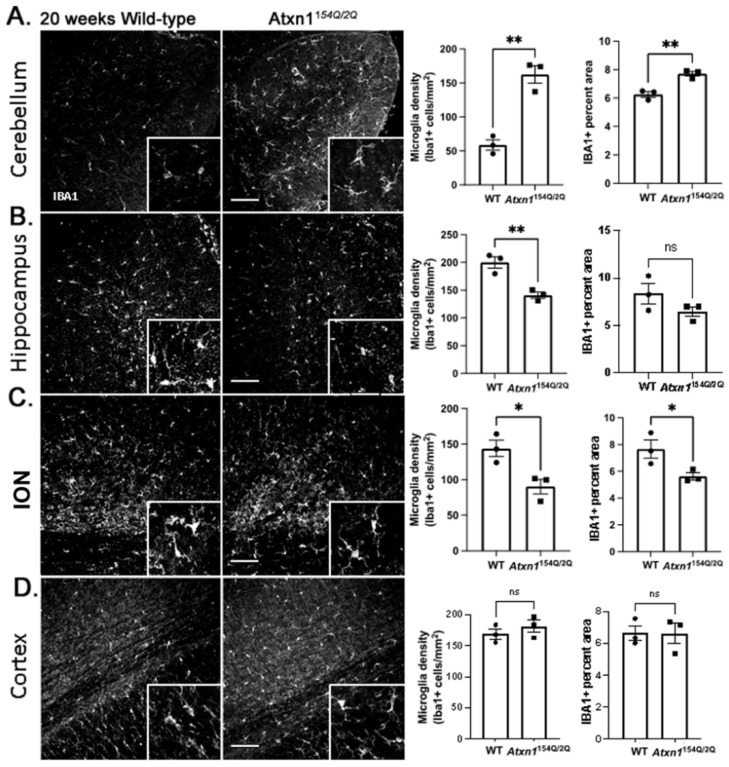
Diversity of late SCA1 microglial morphology phenotypes in *Atxn1^154Q/2Q^* mice. Brain sections from 20 weeks old *Atxn1^154Q/2Q^* and wild-type littermate controls (N = 3–5) were stained for Iba1. Confocal images of cerebellum ((**A**), lobule X), hippocampus ((**B**), dentate gyrus), brainstem ((**C**), ION), and motor cortex ((**D**), layer 6 and corpus callosum) were used to quantify density of Iba1-positive microglia and Iba1 area. Images were acquired at 20× (636.16 µm × 636.16 µm). Scale bars show 100 µm. Insets in the lower right corners of each image show zoomed images (79.5 µm × 79.5 µm) for qualitative assessment of microglia morphology. Data are presented as mean ± SEM with average values for each mouse represented by a dot. N = 3–5 mice of each genotype (WT and *Atxn1^154Q/2Q^*). ns: not significant, * *p* < 0.05, ** *p* < 0.01. Students *t*-test.

**Figure 7 cells-11-03323-f007:**
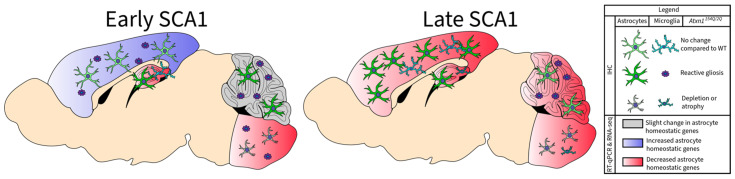
Simplified schematics of astrocytic changes in SCA1. Simplified schematic of astrocytic and microglia changes in SCA1. In early stages of SCA1 (left), brain regions exhibit an increase (cortex) or slight change (cerebellum) in the expression of astrocyte homeostatic genes, whereas regions in red exhibit loss of astrocyte homeostatic genes (hippocampus and medulla). Areas differ in patterns of reactive astrocytes, showing astrocyte atrophy (inferior olivary nucleus (ION)), hypertrophy (cerebellum), or no significant change relative to WT controls (hippocampus and cortex). Most brain regions (cerebellum, ION, cortex) show reactive microglia during early stages of SCA1. During late stages of SCA1 (right), all brain regions exhibit reduced expression of core genes necessary for astrocyte homeostatic function (cortex, cerebellum, hippocampus, and medulla). Most brain regions (cerebellum, hippocampus, and cortex) show reactive astrocytes during late stage of SCA1, although the ION continues to show significant reduction or atrophy of astrocytes. Reactive microglia persist in the cerebellum but are no longer present in the cortex in late SCA1. Some brain regions also show significantly fewer microglia (hippocampus and ION) relative to WT in late SCA1.

**Table 1 cells-11-03323-t001:** Summary of glial changes in SCA1 across brain regions and disease stages.

		Cerebellum	Hippocampus	Brainstem	Cortex
Astrocyte reactivity	Early	Increased ↑ GFAP intensity ↑ GFAP %area	Increased ↑ GFAP intensity ↑ GFAP %area - Astrocyte density	Decreased ↓ GFAP intensity - GFAP %area ↓ Astrocyte density	Not significant - GFAP intensity - GFAP %area - Astrocyte density
	Late	Increased ↑ GFAP intensity ↑ GFAP %area	Increased - GFAP intensity ↑ GFAP %area ↑ Astrocyte density	Decreased - GFAP intensity ↓ GFAP %area ↓ Astrocyte density	Increased ↑ GFAP intensity ↑ GFAP %area ↑ Astrocyte density
Astrocyte homeostasis	Early	Not significant *- Aqp4* *- Gja1* *- Kcnj10* *- Slc1a2*	Decreased *↓ Aqp4* *↓ Gja1* *↓ Kcnj10* *↓ Slc1a2*	Decreased *↓ Aqp4* *↓ Gja1* *↓ Kcnj10* *- Slc1a2*	Increased *↑ Aqp4* *↑ Gja1* *↑ Kcnj10* *- Slc1a3*
	Late	Decreased *↓ Aqp4* *↓ Gja1* *↓ Kcnj10* *↓ Slc1a2*	Decreased *↓ Aqp4* *↓ Gja1* *↓ Kcnj10* *↓ Slc1a2*	Decreased *↓ Aqp4* *↓ Gja1* *↓ Kcnj10* *↓ Slc1a2*	Decreased *↓ Aqp4* *↓ Gja1* *↓ Kcnj10* *↓ Slc1a2*
Microglia reactivity	Early	Increased ↑ IBA1 %area ↑ Microglia density	Not significant - IBA1 %area - Microglia density	Increased - IBA1 %area ↑ Microglia density	Increased ↑ IBA1 %area ↑ Microglia density
	Late	Increased ↑ IBA1 %area ↑ Microglia density	Decreased - IBA1 %area ↓ Microglia density	Decreased ↓ IBA1 %area ↓ Microglia density	Not significant - IBA1 %area - Microglia density

## Data Availability

Data available on request from the authors.

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
