# Peer review of "Spatial and Temporal Diversity of Astrocyte Phenotypes in Spinocerebellar Ataxia Type 1 Mice"

_cells, 2022, doi:10.3390/cells11203323_

Round 1

Reviewer 1 Report

In this study, Rosa et al. studied changes in astrocyte density, expression levels of four astrocyte-specific genes, and microglia density in four different brain regions from early and late symptomatic SCA1 KI mice. In early stage (12 week-old) SCA1 mice, four brain regions showed divergent profiles. In late stage (20 week-old) SCA1 mice, the cerebellum, hippocampus and cerebral cortex showed similar tendency: increase in astrocyte density and decrease in expression levels of four astrocyte-specific genes, suggesting progress of astrogliosis. Thus, local inflammation progresses in the cerebellum, hippocampus and cerebral cortex, along with reactive gliosis in early stage SCA1 KI mice, but likely ameliorates, (but glial scar remains) in late stage SCA1 KI mice.

Microglia density initially increases in response to the local inflammation by reactive gliosis, but returns to the basal level or decreases as the inflammation is mitigated. Therefore, microglia density represents the local inflammation status. In late stage SCA1 KI mice, high microglial density was observed only in the cerebellum, whereas basal or low microglial density was detected in the hippocampus, brainstem and cortex (basal level).

Major points

1.    I recommend that the authors make Table, which summarizes change in astrocyte density, expression levels of astrocyte-specific genes, and microglia density in four different brain regions from both early and late symptomatic SCA1 KI mice. Indeed, I made such Tables by myself to grasp the overall tendency.

2.     

Line 392-396. treatment with Atxn1 targeting antisense oligonucleotides (ASO), which are also taken up by astrocytes, was shown to rescue lower Ins levels in the brainstem and prolong the survival of Atxn1154Q/2Q mice. Based on these results we propose that reduced expression of homeostatic genes and decreased astrocyte viability may contribute to brainstem dysfunction and premature lethality in SCA1 mice.

I think that this part is too speculative. The authors studied 20 week-old SCA1 KI mice, which showed reduced expression of homeostatic genes and maybe decreased astrocyte viability, but SCA1 KI mice at around 20 weeks of age are still alive. SCA1 KI mice normally start to die after 40 weeks of age, after appearance of extensive neuronal damage. Thus, authors should be more cautious.

3.    In Fig. 7, upper diagrams of astrocyte, microglia and synapse have little message. Purple microglia is confusing. Line 415 “late terminal stage” 20-week-old SCA1 KI mice are not terminal stage. Line 417 “Microglia activation in different brain regions does not correlate with astrocyte reactivity or homeostatic gene expression. Yellow represents cell-types shown to express the Atxn1 gene in mice and ATXN1 humans.” → Microglia activation probably correlates with astrocyte reactivity. Authors judge astrocyte reactivity from GFAP intensity and astrocyte density, but these factors do not precisely and timely reflect the astrocyte reactivity (there is some time lag). What does “Yellow” mean? I cannot find yellow cells in the figure. “Atxn1 gene in mice and ATXN1 humans” → “in” is missing between ATXN1 and humans?

Minor points

1.    Line 79. motor cortex of 12-and 24- week-old Atxn1154Q/2Q mice

→ They examined 20-week-old mice, not 24-week-old.

2.    Fig. 1A. A graph of [GFAP+ percent area] should be right hand edge to align with other [GFAP+ percent area] graphs.

3.    Fig. 1B. Graphs show no statistically significant difference. But, variation of data points is big, especially in GFAP intensity graph. It seems that statistical difference appears if they increase the experimental number.

4.    Fig. 1D. There are no descriptions of “ns”.

5.    Fig. 2. Why didn’t authors show a similar graph of the cerebellum as those of other brain regions. In addition, I recommend that authors align graphs similar to Fig. 1 (cerebellum, hippocampus, brainstem, and cortex).

Author Response

We thank the reviewers for their helpful comments and suggestions that have improved our manuscript.

Our point to point responses (in blue) to reviewer’s comments (in italics) are bellow.

Reviewer 1

Major points

  1. I recommend that the authors make Table, which summarizes change in astrocyte density, expression levels of astrocyte-specific genes, and microglia density in four different brain regions from both early and late symptomatic SCA1 KI mice. Indeed, I made such Tables by myself to grasp the overall tendency.

Thanks to reviewer’s comments we have included in the revised manuscript Table 1 (below) that summarizes change in astrocyte reactivity, expression of astrocyte homeostasis genes, and microglia density in four different brain regions from both early and late symptomatic SCA1 KI mice.

  1. Line 392-396. treatment with Atxn1 targeting antisense oligonucleotides (ASO), which are also taken up by astrocytes, was shown to rescue lower Ins levels in the brainstem and prolong the survival of Atxn1154Q/2Q mice. Based on these results we propose that reduced expression of homeostatic genes and decreased astrocyte viability may contribute to brainstem dysfunction and premature lethality in SCA1 mice. I think that this part is too speculative. The authors studied 20 week-old SCA1 KI mice, which showed reduced expression of homeostatic genes and maybe decreased astrocyte viability, but SCA1 KI mice at around 20 weeks of age are still alive. SCA1 KI mice normally start to die after 40 weeks of age, after appearance of extensive neuronal damage. Thus, authors should be more cautious.

We have revised this paragraph to state:

“Based on these and our results we speculate that reduced expression of astrocyte homeostatic genes and decreased astrocyte viability may contribute to brainstem dysfunction and perhaps even to premature lethality in SCA1 mice.”

In our colony the SCA1 Ki mice on average die around 26-28 weeks of age.

  1. In Fig. 7, upper diagrams of astrocyte, microglia and synapse have little message.

We apologize for the lack of clarity; assuming that reviewer refers to little yellow triangles that were meant to signify ATXN1 expression in different cell types. We have revised figure 7 and new Fig 7 does not include yellow triangles.

  1. Purple microglia is confusing.

We have revised Fig 7 and in new Fig 7 microglia are not purple colored.

  1. Line 415 “late terminal stage” → 20-week-old SCA1 KI mice are not terminal stage.

We have revised this to: “late stages”.

  1. Line 417 “Microglia activation in different brain regions does not correlate with astrocyte reactivity or homeostatic gene expression. Yellow represents cell-types shown to express the Atxn1 gene in mice and ATXN1 humans.” → Microglia activation probably correlates with astrocyte reactivity. Authors judge astrocyte reactivity from GFAP intensity and astrocyte density, but these factors do not precisely and timely reflect the astrocyte reactivity (there is some time lag). What does “Yellow” mean? I cannot find yellow cells in the figure. “Atxn1 gene in mice and ATXN1 humans” → “in” is missing between ATXN1 and humans?

We have revised this figure and new figure legend states: “. Simplified schematic of astrocytic and microglia changes in SCA1. In early stages of SCA1 (left), brain regions show exhibit an increase (cortex) or slight change (cerebellum) in the expression of astrocyte homeostatic genes whereas regions in red exhibit loss of astrocyte homeostatic genes (hippocampus and medulla oblongata). Areas differ in patterns of reactive astrocytes, showing astrocyte atrophy (medulla), hypertrophy (cerebellum), or no significant change relative to WT controls (hippocampus and cortex). Most brain regions (cerebellum, medulla oblongata, cortex) show reactive microglia during early stages of SCA1. During late, terminal stages of SCA1 (right), all brain regions exhibit reduced expression of core genes necessary for astrocyte homeostatic function (cortex, cerebellum, hippocampus, and medulla oblongata). Most brain regions (cerebellum, hippocampus, cortex) show reactive astrocytes during late stage of SCA1, although the medulla oblongata continues to show significant reduction or atrophy of astrocytes. Reactive microglia persist in the cerebellum but are no longer present in cortex in late SCA1. Some brain regions also show significantly fewer microglia (hippocampus and medulla oblongata) relative to WT in late SCA1.“

Minor points

  1. Line 79. motor cortex of 12-and 24- week-old Atxn1154Q/2Qmice

→ They examined 20-week-old mice, not 24-week-old.

We appreciate reviewer pointing to this mistake. It has been corrected in the revised manuscript.

  1. Fig. 1A. A graph of [GFAP+ percent area] should be right hand edge to align with other [GFAP+ percent area] graphs.

We have aligned all GFAP+ percent area graphs in revised Figure 1.

  1. Fig. 1B. Graphs show no statistically significant difference. But, variation of data points is big, especially in GFAP intensity graph. It seems that statistical difference appears if they increase the experimental number.

Thanks to reviewer’s suggestions we have performed additional experiments to increase the number of mice and we have indeed now reached statistical significance as shown in revised Figure 1.

  1. Fig. 1D. There are no descriptions of “ns”.

We added Ns to Figure 1D.

  1. Fig. 2. Why didn’t authors show a similar graph of the cerebellum as those of other brain regions.

 In addition, I recommend that authors align graphs similar to Fig. 1 (cerebellum, hippocampus, brainstem, and cortex).

Originally, we decided on correlation graphs for cerebellum instead of a similar graph because of the large variability of data points and interesting high correlation with Gfap expression levels. Based on reviewer’s comments we revised Fig. 2 as reviewer suggested by aligning graphs similar to Fig.1 and showing a similar graph for cerebellum as for the other brain regions. We included correlation graphs from previous Figure 2D as a Supplementary Figure 2.

Reviewer 2 Report

In this article entitled “Spatial and temporal diversity of astrocyte phenotypes in Spinocerebellar ataxia type 1 mice”, the authors find brain region-specific variations that exacerbate with time in astrocytic density, GFAP expression, and markers of astrocytic function by qPCR expression in SCA1 mice. This finding of regional vulnerability is supported by the recent literature on diverse brain regions affected in SCA1. This manuscript further highlights the potential role or reaction of astrocytes to the regional difference caused by mutant ATXN1. 

Overall, this article emphasizes interesting biology that could be further explored in future SCA1 studies of pathomechanism. Only minor comments were noted below that reduced enthusiasm for the manuscript:

1.     Authors note future studies to assess cell autonomy of astrocytic dysfunction could be assessed by conditional depletion of the mutant gene in the cells. However, the converse study of conditional overexpression only in astrocytes should also be discussed.

2.     Scale bars are missing from all IHC figures.

3.     Figure 5 graphical representation of the RNAseq data does not allow for information on variation among mice.

4.     Figure 7 only highlights differences seen in astrocytes however the manuscript presents data toward microglia that show different patterns as noted in the text. These differences could be highlighted in this final figure.

Author Response

We thank the reviewers for their helpful comments and suggestions that have improved our manuscript.

Our point to point responses (in blue) to reviewer’s comments (in italics) are bellow.

Reviewer 2

  1. Authors note future studies to assess cell autonomy of astrocytic dysfunction could be assessed by conditional depletion of the mutant gene in the cells. However, the converse study of conditional overexpression only in astrocytes should also be discussed.

We thank reviewer for this excellent suggestion. Revised manuscript now states: “Future studies using conditional SCA1 mice to selectively express or delete mutant ATXN1 in astrocytes will provide insight into the cell autonomous effect of mutant ATXN1 on astrocyte pathology in the medulla and other brain regions in SCA1”

  1. Scale bars are missing from all IHC figures.

We have added scale bars to all IHC figures.

  1.    Figure 5 graphical representation of the RNAseq data does not allow for information on variation among mice.

We have revised Figure 5 with new Figure 5 showing a heatmap of log2FC and False Discovery Rate p value (FDR) in the brackets that accounts for the individual variation among animals. Since the goal of this figure is to evaluate the differences in relative expression (by Log2FC) between SCA1 and healthy mice, it's not possible to plot individual mice because of how the RNAseq analysis is performed. The goal of differential gene expression analysis is to start with individual data showing gene expression from each mouse and to compare relative expression of each gene between genotypes, accounting for individual variation among mice and among genes. So, the output is the log2FC and FDR (adjusted p) values which doesn't include any data for individual animals. The software assesses variation in the raw read count values within and across genotypes and this is in a big part what determines if they are significantly different or not.

  1. Figure 7 only highlights differences seen in astrocytes however the manuscript presents data toward microglia that show different patterns as noted in the text. These differences could be highlighted in this final figure.

We have revised Figure 7 and new Figure 7 now highlights the differences in both astrocytes and microglia.

Round 2

Reviewer 1 Report

Authors responded my concerns and revised the manuscript. 

I have no further comments. 

Author Response

We thank the reviewer for stating that we have responded to all concerns in revised manuscript and that there are no further comments.